# The effect of caffeine on tinnitus: Randomized triple-blind placebo-controlled clinical trial

Alleluia Lima Losno Ledesma[1], Daniele Leite Rodrigues[2], Isabella Monteiro de Castro Silva[3‡], Carlos Augusto Oliveira[1‡], Fayez Bahmad, Jr[1,2‡]*

1 Postgraduate Program in Health Sciences at the Faculty of Health Sciences, University of Brasília, Brasília, DF, Brazil, 2 Institute of Otorhinolaryngology, Brasília, DF, Brazil, 3 Faculty of Speech Therapy, University of Brasília, Brasília, DF, Brazil

☯ These authors contributed equally to this work.
‡ IMCS, CAO and FB also contributed equally to this work.
* fayezbjr@gmail.com

**Data Availability Statement:** All data presented in the studies were deposited in an appropriate public repository (DOI: 10.6084/m9.figshare.12962684).

## Abstract

### Objective

To test the hypothesis that caffeine can influence tinnitus, we recruited 80 patients with chronic tinnitus and randomly allocated them into two groups (caffeine and placebo) to analyze the self-perception of tinnitus symptoms after caffeine consumption, assuming that this is an adequate sample for generalization.

### Methods

The participants were randomized into two groups: one group was administered a 300-mg capsule of caffeine, and the other group was given a placebo capsule (cornstarch). A diet that restricted caffeine consumption for 24 hours was implemented. The participants answered questionnaires (the Tinnitus Handicap Inventory—THI, the Visual Analog Scale—VAS, the profile of mood state—POMS) and underwent examinations (tonal and high frequency audiometry, acufenometry (frequency measure; intensity measure and the minimum level of tinnitus masking), transient otoacoustic emissions—TEOAE and distortion product otoacoustic emissions—DPOAE assessments) at two timepoints: at baseline and after capsule ingestion.

### Results

There was a significant change in mood (measured by the POMS) after caffeine consumption. The THI and VAS scores were improved at the second timepoint in both groups. The audiometry assessment showed a significant difference in some frequencies between baseline and follow-up measurements in both groups, but these differences were not clinically relevant. Similar findings were observed for the amplitude and signal-to-noise ratio in the TEOAE and DPOAE measurements.

### Conclusions

Caffeine (300 mg) did not significantly alter the psychoacoustic measures, electroacoustic measures or the tinnitus-related degree of discomfort.

URL: https://figshare.com/articles/dataset/
Caffeine_and_Tinnitus_xlsx/12962684.

**Funding:** The authors received no specific funding for this work.

**Competing interests:** The authors have declared that no competing interests exist.

## Introduction

Tinnitus is defined as an auditory perception in the absence of an external sound source. An incidence rate of 25.0 new cases of tinnitus per 10,000 person-years in the United Kingdom has been estimated. In the United States, the prevalence of tinnitus among adults is estimated to range from 10% to 15% [1, 2]. The annoyance of the tinnitus was a result of the tinnitus characteristics and the psychological make up of each individual patient [3] and can affect all aspects of life: it can cause personal, professional, social and family issues, and it can have negative impacts on concentration, sleep, reasoning, memory, emotional balance and social life [4, 5].

The American Academy of Otorhinolaryngology recommends that only patients with tinnitus-related discomfort be identified and treated. For this purpose, it recommends the application of a validated questionnaire; the THI is the most used questionnaire across all regions [1, 6, 7]. The visual analog scale is widely used in clinical practice and research because it is a quick and easy to understand tool, although it has not yet been specifically validated to assess tinnitus [8, 9]. Tinnitus, as a subjective sensation, is directly influenced by the individual's mood [8]. The POMS is the most sensitive questionnaire to assess the effects of caffeine on mood [10].

There are reports of recommendations for performing psychoacoustic measures to assess tinnitus dating back to 1931; however, there is still no consensus on its realization and indication [9]. The European multidisciplinary guideline for the assessment, diagnosis and treatment of tinnitus recommends researching the pitch and loudness of tinnitus as part of the minimum patient assessment [7]. Some authors suggest that this method is an important tool in the characterization of tinnitus and documentation of the efficacy of the adopted therapy [8, 9, 11].

An interruption in the use of caffeine is indicated as part of the treatment for tinnitus. However, the efficacy of this generic treatment lacks scientific evidence [12]. Caffeine is the most commonly consumed psychoactive substance in the world and is found in many products, such as coffee, tea, chocolate, soft drinks, mate herb, powdered guarana, weight loss drugs, diuretics, stimulants, analgesics and anti-allergens. This discontinuation of caffeine use can lead to caffeine withdrawal syndrome, exacerbating symptoms such as headache, irritability and depressed mood [13–15].

Therefore, the present study sought to analyze the influence of caffeine on tinnitus-related discomfort as measured by acoustic, electroacoustic and psychoacoustic assessments. We hypothesize that caffeine use does not change tinnitus-related discomfort.

## Methods

### Study design, approvals, registration and patient consents

This was a randomized, triple blind, placebo-controlled clinical trial approved by the Institutional Research Board of the faculty of health sciences of the University of Brasília (number: 2.031.285) and registered on the national platform for registration of experimental studies—REBEC Platform (RBR-4CRF4D). All data presented in the studies were deposited in an appropriate public repository (DOI: 10.6084/m9.figshare.12962684). The authors confirm that all ongoing and related trials for this drug/intervention are registered.

### Inclusion and exclusion criteria

The sample consisted of individuals who came to the doctor's office reporting tinnitus as the main complaint, of both sexes who were over 18 years of age. Participants who reported objective intermittent or acute tinnitus (less than six months of tinnitus) during a medical

consultation [1], those with psychiatric and/or cognitive disorders that prevented the understanding of tests and questionnaires, and those who used illicit drugs were excluded.

## Recruitment

Four otolaryngologists from a private otorhinolaryngology institute located in Brazil participated in the recruitment process. All individuals who came to the doctor's office of this institute with a main complaint of tinnitus between July 2018 and January 2019 were invited to participate in the study. They were informed about the objectives of the study and completed an anamnesis to report the main characteristics of the tinnitus and to provide their contact information. Subsequently, the main researcher contacted them to inform them about all the research phases and to schedule their participation after obtaining written informed consent.

## Randomization and blinding

Participants were randomly assigned to receive a placebo or a caffeine capsule in a 1:1 ratio by an independent contributor. A completely randomized design was used for both treatments using RANDOMIZATION.COM, with the "First generator" plan that creates random permutations for the treatments used in research. The same employee placed the appropriate capsules in identical packages marked with continuous numbers. The active ingredient (300 mg caffeine) and placebo (cornstarch) were packed in capsules that were identical in color, size and weight. The evaluator, participants and statistician were blinded to the intervention and groups.

## The active ingredient

Caffeine (1, 3, 7-trimethylxanthine) is a central nervous system stimulant belonging to the group of methylxanthines, being fat-soluble and capable of overcoming all biological barriers [16]. It has rapid absorption and 99% is absorbed 45 minutes after ingestion. Plasma concentration in humans is reached 45 to 120 minutes after ingestion and has a half-life of 2.5 to 4.5 hours [17, 18].

Moderate caffeine consumption is classified between 100 and 300mg of caffeine / day, and this consumption is frequently found in individuals around the world [16, 19–21]. Studies suggest that the effects of caffeine start to be observed from 200mg and the toxic effects from an intake of 400mg in healthy adults [22, 23], with reports of previous controlled studies that used doses of 300mg of caffeine [24–26]. Thus, the dosage of 300mg was chosen because it is common in consumers around the world, being pharmacologically active without, however, causing toxic effects.

## Procedures

The research participants first completed a clinical anamnesis to assess the characteristics of their tinnitus and a questionnaire about their eating habits to assess the amount of caffeine they ingested daily.

This study consisted of three phases (Table 1).

A caffeine-free diet was implemented 24 hours before testing. The diet was instructed via telephone contact, providing a detailed list of products that should be avoided via text message or e-mail. The three phases took place on the same day, with an average duration of 4 to 5 hours. In phase 1, the individuals completed the questionnaires and underwent the assessments in the order shown in Table 1. Afterwards, they received the caffeine capsule or the placebo (phase 2). Phase 3 was performed one hour after the ingestion of the capsule with the

**Table 1. Description of the study phases.**

| Phase 1 | Phase 2 | Phase 3 |
|---|---|---|
| THI | Capsule administration | THI |
| VAS | | VAS |
| POMS | | POMS |
| OAE | | OAE |
| Audiometry | | Audiometry |
| Acuphenometry | | Acuphenometry |

same procedures and order as in phase 1. All phases of the study were carried out at the same otorhinolaryngologist institute in which recruitment occurred.

## Questionnaires and exams

In order to find out the amount of caffeine ingested daily by the participants, the adapted eating habits questionnaire was applied and the parameters of the volume and caffeine content of each product of this study was used [22]. The questionnaire investigates the consumption of coffee, tea, mate herb, chocolate, soda, chocolate powder and food and energy supplements, highlighting the type, quantity, brand, shape and frequency of consumption.

To assess the tinnitus-related discomfort, two questionnaires were applied: the tinnitus handicap inventory (THI) and the visual analog scale (VAS). THI seeks to measure the impact of tinnitus on the patient's quality of life. It consists of 25 questions with a total score ranging from 0 to 100. For each item one must answer "yes", "sometimes" and "not" being assigned the score 4 for each "yes" answer, 2 for each answer "sometimes" and 0 for each answer "no". Based on the total score obtained, tinnitus is classified as no handicap (0 to 16 points), mild handicap (18 to 36), moderate handicap (38 to 56), severe handicap (58 to 76) and catastrophic handicap (78 to 100) [27, 28]. The VAS is a black and white scale used to estimate the magnitude that ranges from 0 to 10 divided into three parts: 0–2 (mild), 3–7 (moderate) and 8–10 (intense) [29].

In addition to tinnitus-related discomfort, the individual's mood at the time of the exams was assessed using the Profile of Mood States (POMS). The questionnaire consists of 65 items that describe feelings, where the subjects must attribute the mentions on a 5-point likert scale: 0—nothing, 1—a little, 2—more or less, 3—a lot and 4—extremely. The results found in each affective state were analyzed: tension-anxiety, depression-discouragement, anger-hostility, vigor-activity, fatigue-inertia and mental confusion-perplexity. In addition, the TMD (Total Mood Disturbance) was calculated, which represents an overview of mood, according to criteria determined by the questionnaire authors [30]. All questionnaires were completed in a silent environment in the presence of the main researcher, who answered any questions the participants had about the assessments. Participants had the necessary time to complete the questionnaires.

Tonal audiometry was performed with the warble stimulus at frequencies of 1, 2, 3, 4, 6, 8, 0.5 and 0.25kHz using the descending-ascending technique. Then, high-frequency audiometry was performed at frequencies of 9, 10, 11.2, 12.5, 14, 16 and 18 kHz. These examinations were performed using the AD 229 audiometer from Interacoustics and the HDA 300 headphones, both of which were calibrated annually. Individuals who had hearing thresholds less than or equal to 25dB in all frequencies of conventional audiometry were classified as normal hearing [31].

In acuphenometry, pitch (frequency measure), loudness (intensity measure) and the minimum level of tinnitus masking were evaluated using the same equipment. To measure the pitch, the pure tone stimulus was used if the patient characterized the tinnitus as a whistle, and the narrow band noise was used if patient characterized the tinnitus as a hiss. Two extreme

frequencies were presented at an intensity of 10 dBSL until the patient identified the sound that most resembled his or her tinnitus. Then, loudness was investigated, and the frequency indicated by the patient was presented from the audiometric threshold until the patient described an intensity similar to his or her tinnitus. A 1-dB scale was used. Finally, the minimum level of tinnitus masking was assessed by presenting white noise in the ear contralateral to the tinnitus (if the tinnitus was unilateral) or in the ear where the tinnitus was less intense (if the tinnitus was bilateral) until the individual no longer noticed the tinnitus. If there was no masking, the research was interrupted at 80 dBHL to avoid exposure to a very high intensity. In all measures, the level of sensation was considered).

The electroacoustic evaluation was performed by researching the transient otoacoustic emissions and distortion product using the Otoread Interacoustic (Denmark) instrument at an intensity of 80 dB with 1000 presentations. The frequency bands of 1.5, 2, 3, 4, 5 and 6kHz were evaluated in the transients, and frequency bands of 1.5, 2, 3, 4, 5 and 6kHz were evaluated in the distortion product bilaterally.

## Statistical analysis

The calculation of the sample size was performed using the estimated variability of the THI variable (range 0 to 100) with standard deviation equal to 2.0 and the capacity to detected 3 points in a difference between groups caffeine and placebo, considering the following parameters: an $\alpha$ of 0.05, a test power (1-$\beta$) of 0.8, and 10% of loss of collection, it was determined that 39 individuals were needed in each group.

The distributions of the variables were analyzed, revealing nonnormal distributions for all analyzed variables. Thus, the data were presented as medians and analyzed using nonparametric tests.

Baseline measurements in the two groups were compared using the Chi-square test. The analysis of quantitative variables was performed using the Mann-Whitney nonparametric test and Wilcoxon signed-rank test. The correlation between variables was assessed using Spearman's correlation coefficient. $P < 0.05$ was considered significant. The analysis was performed using the SAS 9.4 application (SAS Institute, Inc., 1999).

## Results

### Recruitment and randomization

A total of 155 individuals were invited to participate. Of these, 7 did not provide valid contact information, 2 reported having intermittent tinnitus, and 66 declined to participate when contacted; thus, we analyzed data from 80 participants,40 in each group (Fig 1).

Recruitment was stopped when the required sample and a 6-month recruitment period has been reached. All data analyses were performed with 40 individuals from each group.

### Sample characterization

The individual characteristics, characteristics of tinnitus and tinnitus-related discomfort were statistically similar in both groups, as shown in Table 2.

Caffeine consumption in the usual diet was not correlated with age, sex, tinnitus duration, or the initial measurements of THI, VAS and loudness (Table 3).

### Comparison of phases 1 and 3

**Questionnaires measurements.** The assessment of mood through the Profile of Mood States demonstrated a statistically significant difference in the TMD (p<0.001), tension-

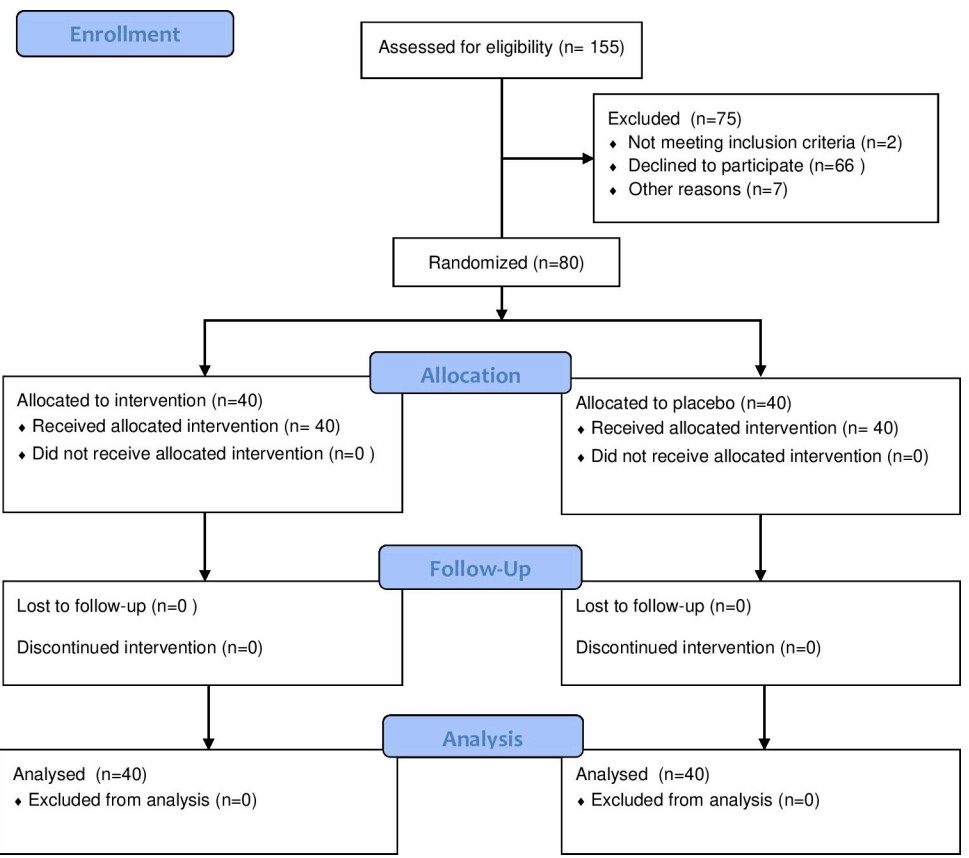

**Fig 1. Flow diagram.**

anxiety (p = 0.009), depression-despondency (p <0.001) and anger-hostility (p = 0.027) states between phase 1 and phase 3 in the caffeine group, with lower values in phase 3 for all states. In the control group, there was a significant difference in the affective states of tension-anxiety (p = 0.026) and mental confusion-perplexity (p = 0.013) between phase 1 and phase 3 with lower values in phase 3.

Regarding the tinnitus-related discomfort measured by the Tinnitus Handicap Inventory (THI), there was a statistically significant difference in THI scores between phases 1 and 3 in both groups, with lower values in phase 3. The caffeine group in phase 1 had a median of 36, IQR of 35.5 and a p-value of 0.001, and the placebo group had a median of 32, IQR of 27.0 and a p-value of <0.001. The analysis of questions 8 and 19 of the THI demonstrated that there was no statistically significant difference between phases 1 and 3. In question 8, a median of 4 p-values of 0.18 was observed in the caffeine group and a median of 4 p-values of 0.79 in the placebo group; in question 19, median of 4 and p value of 0.21 in the caffeine group and median of 4 and p value of 0.97 in the placebo group.

There was no difference in the Visual Analog Scale (VAS) scores between phases 1 and 3 in either group; the caffeine group had a median of 5.0, IQR of 4.0 and a p-value of 0.977, and the placebo group had a median of 6.0, IQR of 5.0 and a p-value of 0.699.

The comparison between the differences found in each group demonstrated that there was no significant difference in any aspect of the questionnaires. When correlating the consumption of caffeine with the results of the questionnaires, there was also no statistically significant difference in any aspect (S1 Appendix).

**Table 2. Intergroup comparison of individual characteristics and tinnitus.**

| | Caffeine group | Placebo group | |
|---|---|---|---|
| | **Percentage** | | **p-value** |
| Gender | 50% Female | 55% Female | 0.654 |
| | 50% Male | 45% Male | |
| Hearing loss | 50% Have | 47.5% Have | 0.502 |
| | 50% Do not have | 52.5 Do not have | |
| Tinnitus type | 82.5% whistle | 75% whistle | 0.412 |
| | 17.5% wheezing | 25% wheezing | |
| Tinnitus location | 50% bilateral | 40% bilateral | 0.693 |
| | 12.5% right ear | 25% right ear | |
| | 12.5% left ear | 17.5% left ear | |
| | 20% In the head | 17.5% In the head | |
| THI classification | 27% No handicap | 22% No handicap | 0.255 |
| | 25% Mild handicap | 37% Mild handicap | |
| | 33% Moderate handicap | 23% Moderate handicap | |
| | 15% Severe handicap | 10% Severe handicap | |
| | 0% Catastrophic handicap | 8% Catastrophic handicap | |
| | **Median (IQR)** | | **p-value** |
| Age (yrs) | 52 (42.25–60) | 52 (43–59,5) | 0.855 |
| Tinnitus duration (mos) | 36 (11.25–147) | 48 (17.25–165) | 0.497 |
| Caffeine consumption (mg/day) | 91.5 (28.5–175.75) | 92 (43.5–200.75) | 0.795 |
| Tritonal mean Right ear (dB) | 16.7 (10–25) | 13.3 (8–23) | 0.304 |
| Tritonal mean Left ear (dB) | 15.0 (12–23) | 14.2 (10–17.75) | 0.429 |
| THI total score | 36 (4–74) | 32 (4–82) | 0.603 |
| VAS | 5 (3–7) | 6 (2–7) | 0.645 |
| POMS (TMD) | -9 (-19–12.5) | -8 (-19–15.75) | 0.864 |

Note: * calculation of p-value of the nonparametric Mann-Whitney test

IQR: Interquartile ranges.

**Psychoacoustic measurements.** The evaluation of pitch revealed a high prevalence of individuals with high-frequency tinnitus and substantial differences in the measurements between phases 1 and 3. There was no difference in loudness or minimal masking level (MML) between phases 1 and 3 in both groups. The caffeine group had a median loudness value and a minimum masking level of 6 dB and 15.5 dB, respectively; the placebo group had a median loudness value and a minimum masking level of 5 dB and 15 dB, respectively.

The caffeine group showed a statistically significant difference between thresholds at frequencies of 2, 3 and 4kHz in the right ear. The placebo group showed a significant difference between thresholds at frequencies of 0.25, 3 and 4kHz in the right ear and at frequencies of 0.25, 0.5, 1, 2 and 3kHz in the left ear. High-frequency audiometry did not reveal difference at any frequency in the caffeine group; in the placebo group, there were significant differences at 18 kHz in the right ear and at 14 kHz in the left ear.

The comparison between the differences found in each group demonstrated that had a statistically significant difference in the thresholds only of the conventional audiometry in the left ear in some frequencies as shown in S1 Appendix. A very weak correlation was found between caffeine consumption and the thresholds in the frequency of 8 and 14kHz in phase 3 in the

**Table 3. Comparison of caffeine consumption in the usual diet across characteristics.**

| Variables | Value | p-value* |
|---|---|---|
| *Age* | | |
| Correlation | 0.15 | 0.181 |
| *Gender*** | | |
| Male | 60 (11,75–168) | 0.378 |
| Female | 37.5 (14,5–120) | |
| *Tinnitus duration* | | |
| Correlation | 0.07 | 0.544 |
| *THI* | | |
| Correlation | -0.02 | 0.846 |
| *VAS* | | |
| Correlation | -0.04 | 0.755 |
| *Loudness* | | |
| Correlation | 0.07 | 0.569 |

Note: *p-value of Spearman's correlation test and

** calculation of median (interquartile ranges)

right ear and 0.5 and 14kHz in phase 1 and 14 and 18kHz in phase 3 in the left ear, and the higher the thresholds the higher the habitual consumption of caffeine.

**Electroacoustic measurements.** For distortion product otoacoustic emission (DPOAE) assessments, the caffeine group showed a statistically significant difference in the signal-to-noise ratio at the frequency of 3kHz in the left ear between phases 1 and 3, with higher ratios in phase 3. The placebo group showed significant differences in amplitude at 4kHz and in the signal-to-noise ratio at 3kHz in the right ear (Fig 2).

For the transient otoacoustic emission (TEOAE) assessments, the caffeine group showed a statistically significant difference in the signal-to-noise ratio at the frequency of 3.5kHz in the right ear and at 4kHz in the left ear between phases 1 and 3, with higher values in phase 3. The placebo group showed differences in amplitude at 4kHz in the right ear and in the signal-to-noise ratio at 3kHz in the left ear (Fig 2).

The comparison between the differences found in each group demonstrated a statistically significant difference in the signal-to-noise ratio in the left ear at EOA-PD at 3kHz in phases 1 and 3 and at EOA-T at 3.5kHz in phases 1 and 3 and at 4kHz in phase 1. No correlation was found between caffeine consumption and the parameters analyzed in otoacoustic emissions (S1 Appendix).

**Harms.** When asked about any change in sensation between phases 1 and 3, 74 individuals reported not having noticed any change. Of those who reported change, 4 individuals reported adverse effects, such as palpitation (placebo group), "pressure in the ear" (caffeine group), "headache" (placebo group) and "pressure in the head" (placebo group). One individuals in the caffeine group reported "being more alert", and another individual from the same group reported a lack of headache.

## Discussion

The sample profile of the present study is similar to that previously reported in the literature, being representative of this population. The individual characteristics of the sample were similar regarding the number of males and females and an average age of approximately 50 years [8, 32, 33]. The duration of tinnitus was shorter than previously reported [8, 32], and the location of the

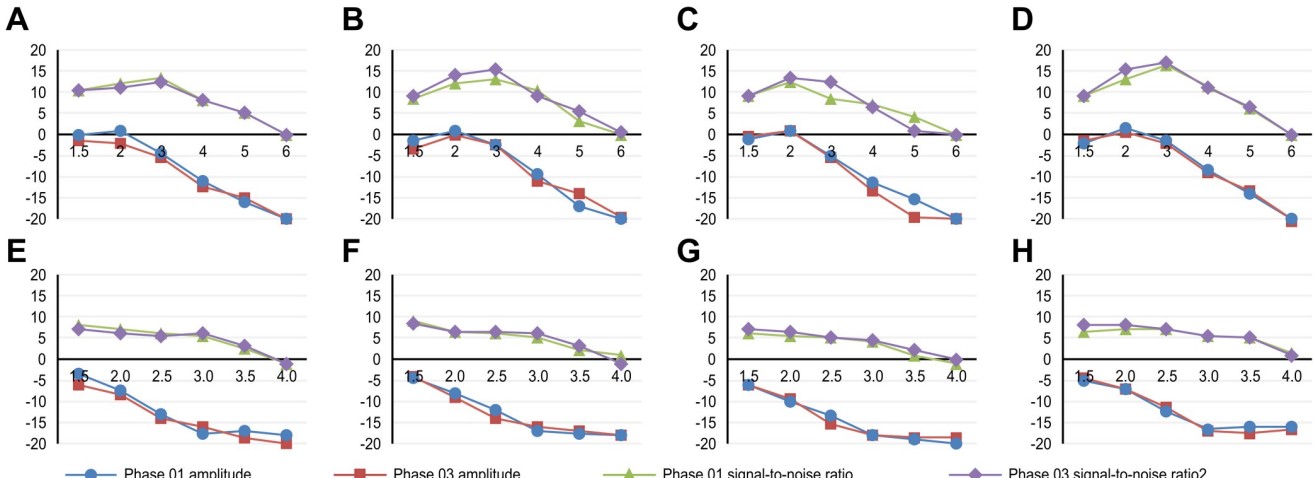

**Fig 2. Parameters in distortion products and transient otoacoustic emissions in the caffeine and placebo groups.** A- DPOAE for the right ear in the caffeine group; B- DPOAE for the right ear in the placebo group; C- DPOAE for the left ear in the caffeine group; D- DPOAE for the left ear in the placebo group; E- TEOAE for the right ear in the caffeine group; F- TEOAE for the right ear in the placebo group; G- TEOAE for the left ear in the caffeine group; H- TEOAE for the left ear in the placebo group.

tinnitus had a similar distribution to that reported in the literature, being more prevalent bilaterally [32, 34, 35]. The THI and VAS values were similar to those previously reported [9, 36].

Hearing loss is believed to be a risk factor for tinnitus [37]. In the present study, a high number of individuals had normal hearing or hearing loss at isolated higher frequencies. A reduction in thresholds was observed at higher frequencies, indicating a cochlear lesion at baseline even in individuals with normal hearing [32].

The median caffeine consumption was 91.5 mg/day in the caffeine group and 92 mg/day in the placebo group, which is considered to be a low level of consumption. Studies using the POMS have shown an increase in TMD [26, 38] and in vigor-activity state [39–41] and a reduction in fatigue-inertia [39–42] and in mental confusion-bewilderment [26, 38], after caffeine consumption. The present study showed an improvement in mood after caffeine consumption and a significant reduction in the tension-anxiety, depression-discouragement and anger-hostility states. The lack of an increase in vigor activity may be associated to the low daily consumption of caffeine among the sample, since an increase in this parameter has only been shown in habitual caffeine consumers [43].

When comparing the results of phases 1 and 3, were observed changes in THI and in all tests performed in both groups: caffeine and placebo. This fact suggests that the effects on tinnitus cannot be directly linked to caffeine, since in the placebo group, there were also changes identified after taking caffeine-free capsules. THI questions 8 and 19 were analyzed individually because, although there is a recommendation in the literature on the use of the total score in research and clinical practice [44], it was demonstrated that these questions appear to be more sensitive to the effects of changes [11, 45]. The non-significant variation between the values of phases 1 and 3 in both groups corroborate the hypothesis that caffeine does not influence the discomfort with tinnitus.

The influence of caffeine on tinnitus-related discomfort has been investigated by other researchers using different instruments. A double-blind pseudorandomized clinical trial in the United Kingdom used four analysis instruments and found that caffeine abstinence did not improve tinnitus-related discomfort. The authors added that caffeine abstinence led to side effects such as headache and nausea [12].

Only two studies that investigated the relationship between caffeine and tinnitus using the THI and VAS instruments. One of them revealed no correlation between VAS scores and caffeine consumption in the usual diet as observed in the present sample however, because it is a retrospective cross-sectional study, it was not proposed to investigate the results of VAS after consuming a certain dose of caffeine. [46]. The other study evaluated the influence of caffeine reduction in the long term (1 month) on VAS and THI results, finding a positive relationship in them; however, that study was not blinded, which may have influenced the patient's responses to the questionnaire [36].

In the present study, a high prevalence of individuals with high-frequency tinnitus was observed. These data diverged from a previous study where it was found that 68% of the individuals had tinnitus with a pitch of up to 3000Hz [34] and another that reported 6kHz to be the most prevalent pitch [9]. These studies describe the pitch of tinnitus in adult individuals without relating them to caffeine consumption. It is worth mentioning that the mentioned studies do not research pitch at high frequencies. Herein, there was great variability in pitches between phase 1 and 3 measurements in both groups. This low replicability between measurements at different times was noted earlier, suggesting the need for the test to be repeated more times to learn the procedure [35, 47].

The loudness found in the sample was similar to those previously reported, which were up to 3 dB in 61% of individuals and between 4 and 6 dB in 28% of individuals [34]. It has been previously demonstrated that the loudness measured at the frequency of tinnitus presents only a few decibels above thresholds due to the occurrence of recruitment. An abnormally loudness growth occurs leading to the sound being perceived as much higher than their low sensation levels would indicate [47, 48].

There was no statistically significant difference in loudness between phases 1 and 3, suggesting that caffeine had little influence on loudness. It is worth mentioning that although the replicability rate of tinnitus psychoacoustic measurements is variable, loudness tends to have better replicability, with variations of no more than 2dB observed when measurements are replicated on the same day [47].

In both groups, a statistically significant difference was found in the audiometric thresholds between phases 1 and 3 for some frequencies in both conventional and high frequency audiometry. It is known that pure tone audiometry is influenced by endogenous factors such as attention and patient experience [49]; the difference observed herein was not of clinical value since only two individuals in each group showed variation in the degree of hearing loss grade between phases 1 and 3, and this variation in degree was due to the difference of less than 10dB between the measures which is considered a non-significant variation [50]. These data differ from out expectations since caffeine has been shown to improve an individual's alertness, ability to concentrate, attention and memory and improves transmission in the central brain auditory pathways [51, 52].

The electroacoustic evaluation showed a difference in the measurements of otoacoustic emissions in terms of amplitude and signal-to-noise ratio at some frequencies in both the caffeine group and the placebo group. The amplitude and signal-to-noise ratio decreased with increasing frequencies. However, it has been observed that the absence of responses at 5kHz and 6kHz is not uncommon in adults without tinnitus even with thresholds within the normal range among these frequencies [53].

The limitations of this study involve measurement bias, as the THI is not meant for measuring momentary variations in the sensation of tinnitus and acuphenometry is influenced by repeated measurements. These methods, despite their limitations, were used because there are no other validated and widely used methods in clinical and scientific practice that met the objectives set.

## Conclusion

Caffeine at a dose of 300 mg did not significantly alter the psychoacoustic measures, electro-acoustic measures or tinnitus-related discomfort among light caffeine consumers.

## Supporting information

**S1 Checklist. CONSORT checklist.**
(PDF)

**S1 Appendix. Difference groups correlation with caffeine consumption.**
(DOCX)

**S2 Appendix. Consubstanced opinion in Portuguese (original).**
(PDF)

**S3 Appendix. Consubstanced opinion in English (translated).**
(DOCX)

## Author Contributions

**Conceptualization:** Alleluia Lima Losno Ledesma, Daniele Leite Rodrigues, Isabella Monteiro de Castro Silva, Carlos Augusto Oliveira, Fayez Bahmad, Jr.

**Data curation:** Alleluia Lima Losno Ledesma, Daniele Leite Rodrigues.

**Investigation:** Alleluia Lima Losno Ledesma, Daniele Leite Rodrigues.

**Methodology:** Alleluia Lima Losno Ledesma, Daniele Leite Rodrigues, Isabella Monteiro de Castro Silva, Carlos Augusto Oliveira, Fayez Bahmad, Jr.

**Project administration:** Isabella Monteiro de Castro Silva, Carlos Augusto Oliveira, Fayez Bahmad, Jr.

**Resources:** Alleluia Lima Losno Ledesma, Daniele Leite Rodrigues, Isabella Monteiro de Castro Silva, Carlos Augusto Oliveira, Fayez Bahmad, Jr.

**Supervision:** Isabella Monteiro de Castro Silva, Fayez Bahmad, Jr.

**Validation:** Alleluia Lima Losno Ledesma, Daniele Leite Rodrigues.

**Visualization:** Alleluia Lima Losno Ledesma, Daniele Leite Rodrigues.

**Writing – original draft:** Alleluia Lima Losno Ledesma, Daniele Leite Rodrigues.

**Writing – review & editing:** Alleluia Lima Losno Ledesma, Daniele Leite Rodrigues, Isabella Monteiro de Castro Silva, Carlos Augusto Oliveira, Fayez Bahmad, Jr.

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
