## [Decision Letter · Decision Letter 0]

31 Dec 2020

PONE-D-20-30851

The effect of caffeine on tinnitus: Randomized triple-blind placebo-controlled clinical trial

PLOS ONE

Dear Dr. Bahmad,

Thank you for submitting your manuscript to PLOS ONE. After careful consideration, we feel that it has merit but does not fully meet PLOS ONE’s publication criteria as it currently stands. Therefore, we invite you to submit a revised version of the manuscript that addresses the points raised during the review process.

We look forward to receiving your revised manuscript.

Kind regards,

Hong-Liang Zhang, M.D., Ph.D.

Academic Editor

PLOS ONE

Journal Requirements:

2. Please provide references for the THI and VAS questionnaires.

3. In your Methods section, please provide additional information about the participant recruitment method and the demographic details of your participants. Please ensure you have provided sufficient details to replicate the analyses such as descriptions of where participants were recruited and where the research took place.

4.Thank you for submitting your clinical trial to PLOS ONE and for providing the name of the registry and the registration number. The information in the registry entry suggests that your trial was registered after patient recruitment began. PLOS ONE strongly encourages authors to register all trials before recruiting the first participant in a study.

1) your reasons for your delay in registering this study (after enrolment of participants started);

2) confirmation that all related trials are registered by stating: “The authors confirm that all ongoing and related trials for this drug/intervention are registered”.

5. Thank you for including your ethics statement:  "This was a randomized, triple blind, placebo-controlled clinical trial approved by the Institutional Research Board (number: 2.031.285) and registered on the national platform for registration of experimental studies - REBEC Platform (RBR-4CRF4D). All data presented in the studies were deposited in an appropriate public repository (DOI: 10.6084 / m9.figshare.12962684).

All participants were informed about the objectives of the study and signed the written informed consent.".   

7. Please include a caption for figure 2.

Reviewers' comments:

Reviewer's Responses to Questions

**Comments to the Author**

1. Is the manuscript technically sound, and do the data support the conclusions?

Reviewer #1: No

2. Has the statistical analysis been performed appropriately and rigorously? 

Reviewer #1: I Don't Know

3. Have the authors made all data underlying the findings in their manuscript fully available?

Reviewer #1: Yes

4. Is the manuscript presented in an intelligible fashion and written in standard English?

Reviewer #1: Yes

5. Review Comments to the Author

Reviewer #1: In this randomized, clinical trial, the authors tested the hypothesis that caffeine can influence tinnitus. Eighty patients with chronic tinnitus were recruited and divided into two groups, one of which received a 300-mg capsule of caffeine, and the other a placebo capsule, after following a caffeine-free diet for 24 hours before testing. Participants completed questionnaires and underwent various assessments before, and one hour following, receipt of the capsule. Tinnitus-related discomfort was assessed using the tinnitus handicap inventory (THI) and the visual analog scale (VAS). Mood was assessed using the Profile of Mood States (POMS), and an electroacoustic evaluation was conducted. Tonal audiometry was performed, and pitch, loudness and minimum level of tinnitus masking were also evaluated.

Although for the most part the authors have done a good job of following the CONSORT guidelines, in some instances, their efforts have not, I feel been sufficient.

Although the authors describe in detail the questionnaires and exams that were administered, it is unclear which of these constitutes the primary outcome on which their sample size calculation was based. This must be clarified. Although it is stated (l.138) that the study is powered for an effect size of 0.5, the authors do not specify to which outcome(s) this applies. In addition to clarifying this information, the authors should justify why this is considered to be a “clinically important” effect.

ll.93-94. Given that the daily amount of caffeine consumption would likely have an impact on the effect of the intervention dose, it would be helpful if the authors were to go into more detail about how they measured this in the participants. As the authors point out, caffeine “is the most commonly consumed psychoactive substance in the world” and is found in many products. Was the questionnaire used to estimate daily caffeine intake previously published and validated? The authors should provide a reference for any materials (surveys, questionnaires, etc.) used to determine caffeine consumption, and perhaps include or summarize the questionnaire in the manuscript. If the authors did not use a validated questionnaire, this should also be discussed and justified.

l.99. Similarly, the authors should provide information as to how the participants were instructed to restrict caffeine consumption for 24 hours preceding the intervention. Were the participants given a list of foods/beverages that they were not to consume?

Given that these constitute important parts of the intervention procedure, they should be described in adequate enough detail to allow replication (CONSORT Item 5).

The trial is very short, with all measurements made on the same day in a short amount of time (with only an hour of time elapsed between ingestion of the capsule and “after” measurements). Could the authors provide some explanation to justify this approach? A brief explanation of the chemical effects of caffeine (e.g., fast-acting? Lasting how long? Peaking at what point?) is warranted here.

Could the authors explain/justify their decision to use a 300mg dose of caffeine? Is this considered a large enough amount to affect habitual consumers of caffeine? What about those who report NO caffeine consumption? It does not appear to have been decided based on the caffeine usage of the participants, nor was the “usual” individual caffeine consumption taken into account in the analysis, although there is a very wide range of use as reported in Table 1.

Medians (Table 1) should be reported with interquartile ranges (IQRs). Ranges are very sensitive to outliers, and do not give an adequate representation of a distribution. IQRs should always be reported instead of, or in addition to, ranges.

Table 2.

• Although the authors describe the tests used for which results are reported here, they do not indicate the range of the possible scores or how these scores are interpreted. This makes their reported results (citing numerical differences) difficult to interpret.

• It is unclear what correlation is being examined. Is it the correlation between estimated daily use prior to the trial?

• It is not clear what is mean by the note for Gender comparison: “calculation of mean (standard deviation) and p-value of the non-parametric Mann-Whitney test.” These values are clearly not normally distributed and should be summarized using medians and IQRs, not means and standard deviations. The authors have appropriately conducted a non-parametric test, while reporting parametric summaries.

l.137. What is the ARE method of sample size calculation? Please provide a reference.

In the Discussion, the authors compare their data and results to a number of other studies. They do not clearly describe, however, why these particular studies are comparable. To say that “the individual characteristics of the sample were similar to those reported in other studies” is not enough; the justification for comparison to these particular studies is needed. Are they also studies of caffeine use and tinnitus? Did they examine interventions of similar dose and duration? The duration of tinnitus is said to be “shorter than previously reported” but does that mean in studies of tinnitus in general? Or in studies of caffeine use in those with tinnitus? It would be helpful if the authors provided enough information about the comparative studies to demonstrate why they should or should not be comparable. E.g., the current study is compared to a study of caffeine abuse (although the participants in the current study are stated to be low consumers) and a study of a month-long reduction of caffeine consumption. These would not appear to be easily comparable to the study reported here.

ll.269-70. Since it is not known why the placebo group improved in THI and VAS scores, the authors should be more cautious in attributing it to “feeling like their tinnitus complaints were being taken seriously.” Similarly, in ll.293-94, the authors state that that the difference observed between the groups “was not clinically significant” (this should be explained) and “can be attributed to the patient’s experience with the exam.” (What does this mean? Perhaps the use of the phrase “might be attributable” would be more cautious here.)

Overall, I found that the discussion section did succeed in logically constructing a story formed by the results of the study. It would benefit from a tighter construction; as it is, it feels like it wanders from one point to another. Some important issues that warrant, but are not given, discussion include the possible differing effects of the caffeine on participants with widely differing caffeine usage, potential difference of effects due to duration of tinnitus, or why such a brief trial would be expected to yield meaningful results. The length and interventions in other trials are not detailed for comparison or justification for the methodology employed here.

The authors point out that a limitation of the study is that they used a measurement (THI) “not meant” for measuring what they purport to measure. Is this a limitation of the study, or of the authors’ design of the study? If they utilized as an outcome measure a scale “not meant for measuring momentary variations in the sensation of tinnitus” and another tool “influenced by repeated measurement,” the use of these less-than-ideal tools should be justified. Did they have no other choices? How can they expect to see results if they use tests that they know cannot adequately detect them?

While the authors are to be commended for conducting a clinical trial largely within the CONSORT guidelines, I feel that a much better job of reporting the justification for their design daecisions, the reasoning behind their methodology and the interpretation of their results compared to other (appropriately chosen) studies ais needed before the study can be published.

6. PLOS authors have the option to publish the peer review history of their article (what does this mean?). If published, this will include your full peer review and any attached files.

Reviewer #1: No

---

## [Author Response · Author response to Decision Letter 0]

21 Jan 2021

Response to Reviewers

I would like to thank the considerations made regarding the submitted manuscript. They were taken into account and certainly added quality to the study. Below are the answers to each consideration made.

• The naming of the files has been formatted according to the journal's guidelines

• The recruitment method as well as the recruitment and data collection location was further detailed in the text as requested: Chapter Methods, subtopics "Recruitment" (page 5, lines 89-90) and "Procedures" (page 7, lines 133-134)

• I would like to apologize for the mistake in the article. There was an error writing the foreign language. Registration was made on the REBEC platform in March, being accepted in July. Recruitment started shortly after being "accepted" in July (not in June as previously described in the article): Chapter Methods, subtopics "Recruitment" (page 5, line 91)

• The requested declaration “The authors confirm that all ongoing and related trials for this drug / intervention are registered” has been added in the chapter "Methods" subtopic "Study design, approvals, Registration and Patient Consents" (page 4, line 78-79).

• The full name of the institutional review board has been added in the "Methods" subsection "Study design, approvals, Registration and Patient Consents" chapter (page 4, line 75), as well as in the submission form

• The corresponding author's ORCID ID was added in Editorial Manager

• The caption for figure 2 has been included: page 9, line 195

• Sample size calculation: The description of the sample size calculation has been reformulated in order to make it clearer (Chapter “Methods” subtopic “Statistical analysis”, page 9, lines 174-178).

o For the sample calculation, the THI results were considered as the primary outcome

o The sample size considering that non-parametric tests were used was determined by multiplying the sample size calculated for parmetric tests by a correction factor. This correction factor is ARE (Asymptotic Relative Efficiency), described by Pitman. To calculate the sample size, α equal to 0.05 was used, test power (1-β) equal to 0.8, effect equal to 0.5, using the conservative ARE minimum method. The ARE method reference was: Lehmannn, EL. Nonparametrics: Statistical methods based on ranks. San Francisco, CA: Holden-Day, 1975; Pitaman, E.J.G, (1948), Lecture notes on nonparametric statistical inference, Columbia University.

• The description of the calculation of the amount of habitual consumption of caffeine has been included, as well as its reference: Chapter “Methods” subtopic “Questionnaires and exams”, page 7, lines 137-141 and reference [20].

• The description of how the participants were instructed to restrict caffeine consumption for 24 hours preceding the intervention has been included: Chapter “Methods” subtopic “Procedures”, page 6/7, lines 128-130.

• The subtopic "The active ingredient" was added in chapter “Methods” in order to explain the choice of measures 1 hour after caffeine consumption based on its chemical effects (page 5/6, line 107-111) and the choice of 300mg dose (page 6, line 112-118). 

• Table 1:

o Range values have been replaced by IQRs (page 10)

• Table 2:

o In order to allow the correct interpretation of the values presented in "Table 2", the description of the tests used (THI and VAS) was added in the chapter "Methods", sub item "Questionnaires and exams" (page 7, lines 143-144), as well as their references [25,26]

o The title of the table has been adjusted to show that it represents the correlation between caffeine consumption in the usual diet and the characteristics described in the same table (page 11)

o An error occurred in the formatting and layout of the "*" signs. They were reorganized and presented median and interquartile ranges (pages 11/12)

• The "Discussion" chapter has been revised to make it more compact and cohesive in order not to wander from one point to another (pages 14-18). Because of this, a revision of the "Introduction" chapter was necessary (page 3)

o Page 14, lines 276-280: The first paragraph of the discussion was structured in order to verify whether our sample was similar to the profile of patients with tinnitus reported in studies conducted in this population previously regarding sex, age, duration and location of tinnitus and discomfort in relation to it (calculated with the same instrument used in this study).

o The studies to which ours was compared were better described, as proposed, in order to make this comparison clearer: page 16, lines 306-312 and 321-323. A comparison was made between the values found in the VAS in our study with the values found in the only two studies that investigated the relationship between caffeine and tinnitus and that used the same instrument. One of these studies was cross-sectional and retrospectively investigated in 107 individuals whether there was a relationship between the discomfort with tinnitus (assessed by VAS) and the consumption of caffeine in the usual diet. This relationship is not observed. The other study assessed the influence of long-term caffeine reduction on VAS results. Our study was compared to previous studies that presented the pitch and loudness of tinnitus in a sample of adult individuals, without, however, relating these measures to caffeine consumption. These studies were chosen as the studies that linked caffeine to tinnitus showed no such measures

o Some inferences have been restructured to make them more cautious, as recommended: page 16, lines 316-317; page 17, lines 336/339

o Observations on the limitations of the study were added in order to justify the choice of the procedures cited for the study methodology: page 18, lines 351-353

---

## [Decision Letter · Decision Letter 1]

10 Mar 2021

PONE-D-20-30851R1

The effect of caffeine on tinnitus: Randomized triple-blind placebo-controlled clinical trial

PLOS ONE

Dear Dr. Bahmad,

Thank you for submitting your manuscript to PLOS ONE. After careful consideration, we feel that it has merit but does not fully meet PLOS ONE’s publication criteria as it currently stands. Therefore, we invite you to submit a revised version of the manuscript that addresses the points raised during the review process.

The author are expected to fully take into account the reviewers' comments and to substantially revise their manuscript.

We look forward to receiving your revised manuscript.

Kind regards,

Hong-Liang Zhang, M.D., Ph.D.

Academic Editor

PLOS ONE

Additional Editor Comments (if provided):

Line 21

Define what the placebo is in the abstract.

Line 26.

24 hours does not seem long as control…? Explain what this is.

Line 27.

VAS here and elsewhere

I think by VAS you mean magnitude estimation. In psychophysics when you are trying to estimate the magnitude of something, it is referred to as Magnitude Estimation. There are several psychophysical procedures that can be used to estimate the magnitude. VAS is one. If one is going to use a VAS scale, it is necessary to state the length of the line, what markings and labels are on the line, and what resolution will be used to convert the subjects marking into a number.

I prefer to have subjects provide a number from 0 to 100.

Line 29

Most readers will not know what acufenometry is.

Line 35

How do you know what is clinically relevant?

Line 48

Several parts of the brain will be involved in the representation of the tinnitus and of the reactions to the tinnitus. Treatments can be focused on reducing the tinnitus (e.g. pills) or on reducing the reactions to the tinnitus (eg. Counseling )

Line 52

The THI actually uses a 3 label category scale. (SS Stevens.. often referred to as the father of human psychophysics)

Please discuss in the results, the sensitivity of the THI has been challenged.

Line 53

VAS

I think by VAS you mean magnitude estimation. In psychophysics when you are trying to estimate the magnitude of something, it is referred to as Magnitude Estimation. There are several psychophysical procedures that can be used to estimate the magnitude. VAS is one. If one is going to use a VAS scale, it is necessary to state the length of the line, what markings and labels are on the line, and what resolution will be used to convert the subjects marking into a number.

I prefer to have subjects provide a number from 0 to 100.

Line 60

Measuring tinnitus

The dB sensation level is not a direct measure of loudness.

They highlighted 4 reasons for measuring tinnitus.

1. Confirm to the patient that their tinnitus is real

2. Monitor changes

3. Provide insights into mechanisms

4. Aid in the fitting of some devices

Line 132

Define the duration of the phases.

Line 140

Mate ??

Line 154

Define the protocols so that some can replicate your study, a fundamental of science.

Line 197

Not clear what the 6 month period stands for.

Line 283

Normal hearing..

State how you define this

Normal pure tone thresholds (=< 25 dBHL)

317. changes over time

Line 320

300 Hz. Are you sure??

Please cite Pan et al., whose pitch matching data suggests that most patients do not have a pitch match frequency just below the maximum hearing threshold loss frequency. The argues against the brain reorganization model in most subgroups of tinnitus.

Line 327

dB is not a scale of loudness

Tinnitus sensation level is not a measure of loudness. You might want to consider referencing:

Line 352

Reviewers' comments:

Reviewer's Responses to Questions

**Comments to the Author**

1. If the authors have adequately addressed your comments raised in a previous round of review and you feel that this manuscript is now acceptable for publication, you may indicate that here to bypass the “Comments to the Author” section, enter your conflict of interest statement in the “Confidential to Editor” section, and submit your "Accept" recommendation.

Reviewer #1: (No Response)

Reviewer #2: (No Response)

2. Is the manuscript technically sound, and do the data support the conclusions?

Reviewer #1: Partly

Reviewer #2: Yes

3. Has the statistical analysis been performed appropriately and rigorously? 

Reviewer #1: No

Reviewer #2: Yes

4. Have the authors made all data underlying the findings in their manuscript fully available?

Reviewer #1: Yes

Reviewer #2: Yes

5. Is the manuscript presented in an intelligible fashion and written in standard English?

Reviewer #1: No

Reviewer #2: Yes

6. Review Comments to the Author

Reviewer #1: Although the authors have improved the paper in response to prior comments, I still feel that a number of points remain to be clarified, as noted below. My primary concern, however, is that all comparisons are made within the two groups (caffeine and placebo) individually, and it does not appear that the two groups were ever compared to one another. In order to determine whether caffeine indeed has an effect, changes in the caffeine group need to be compared head-to-head to changes in the placebo group (before and after intervention). The within group changes are primarily important in terms of their comparison across the two intervention groups (i.e., is there a difference of differences?).

Other comments:

The authors imply in the introduction that the THI is used in the diagnosis of tinnitus; in ll.143-44 they report that the THI has 25 questions with a total score ranging from 0 to 100. However, it is not clear how the scores are interpreted or what constitutes a “change." Are varying degrees of tinnitus defined/categorized from the continuous score? If not, how is the continuous score interpreted, and what constitutes a meaningful change in score? The authors state that the sample size was based on a change of 3 points. This seems like a very small change, and could refer to either a change within severity category, or a change from one severity category to another (if the continuous scores are indeed categorized into severity levels). Could the authors please give more information on this questionnaire, and the interpretation of the scores? This information would be important if the trial were to be replicated.

Could the authors provide more information about the POMS? It appears to be testing different domains; is there an overall score in addition to the individual domain scores?

l. 68: “the influence of caffeine on tinnitus-related discomfort”: the authors should state their hypothesis clearly rather than imply it (e.g., we hypothesize that caffeine use will increase/decrease tinnitus-related discomfort).

ll.82-83: Could the authors be more specific about the eligibility criteria? Clearly the study required not just that individuals “have tinnitus” but that they had experienced it for some length of time and at some level of severity/chronicity. These requirements should be clearly stated.

l.219: What is “TMD” an abbreviation for?

ll. 219-23: I am assuming the numbers reported in parentheses are p-values. To be clear, please report as (p<0.001) rather than (<0.001).

ll.221-23: Were the values in the states of tension-anxiety and mental confusion-perplexity between phase 3 higher or lower than those in phase 1?

ll. 226-27: Are 36 and 33 the medians for the two groups at phase 3 or the median changes from phase 1 to phase 3? Please give IQRs when reporting medians.

ll.229-30: Same question with respect to the VAS. Are these medians or median changes? Please report IQRs.

Table 2 p-values. I believe that the majority of these p-values were derived using Spearman’s Correlation Coefficients (and not the Mann-Whitney test, with the exception of Gender).

I don’t feel that the question of how varying levels of “regular” caffeine consumption might affect the outcome of the study has been addressed, although the authors have added information about the level of the dosage and the effects of caffeine as related to time. Just reporting these levels doesn’t take them into account in the analysis.

I still feel that both the Introduction and Discussion sections need work. Their organization is awkward, and the ideas don't flow easily or logically throughout. I feel that much of the information is there, but it is not well presented or coherently organized. Could the authors perhaps find someone to help them with the actual writing of the manuscript? I feel that while there is interesting information here, it just is not presented well.

Reviewer #2: 

Line 21

Define what the placebo is in the abstract.

Line 26.

24 hours does not seem long as control…?  Explain what this is.

Line 27.

VAS here and elsewhere

I think by VAS you mean magnitude estimation.  In psychophysics when you are trying to estimate the magnitude of something, it is referred to as Magnitude Estimation.   There are several psychophysical procedures that can be used to estimate the magnitude.  VAS is one.  If one is going to use a VAS scale, it is necessary to state the length of the line, what markings and labels are on the line, and what resolution will be used to convert the subjects marking into a  number.

I prefer to have subjects  provide a number from 0 to  100.

Line 29

Most readers will not know what acufenometry is.

Line 35

How do you know what is clinically relevant?

Line 48

Please cite the Psychological Model, proposed by Tyler, Aran and Dauman (1992), suggested that the overall annoyance of the tinnitus was a result of the 1) tinnitus characteristics and the 2) psychological make up of each individual patient.  Several parts of the brain will be involved in the representation of the tinnitus and of the reactions to the tinnitus.  Treatments can be focused on reducing the tinnitus (e.g. pills) or on reducing the reactions to the tinnitus (eg. Counseling )

Tyler, R. S., Aran, J-M., & Dauman, R.  (1992).  Recent advances in tinnitus.  Am J Audiol, 1(4): 36-44.

Line 52

The THI actually uses a 3 label category scale.  (SS Stevens.. often referred to as the father of human psychophysics)

Please discuss in the  results, the sensitivity of the THI has been challenged.

Tyler, R.S., Oleson, J., Noble, W., Coelho, C., Ji, H. (2007). Clinical trials for tinnitus: Study populations, designs, measurement variables, and data analysis. Progress in Brain Research, 166: 499-509.

Line 53

VAS

I think by VAS you mean magnitude estimation.  In psychophysics when you are trying to estimate the magnitude of something, it is referred to as Magnitude Estimation.   There are several psychophysical procedures that can be used to estimate the magnitude.  VAS is one.  If one is going to use a VAS scale, it is necessary to state the length of the line, what markings and labels are on the line, and what resolution will be used to convert the subjects marking into a  number.

I prefer to have subjects  provide a number from 0 to  100.

Line 60

Measuring tinnitus

The dB sensation level is not a direct measure of loudness.  

Please reference

Tyler, R. S. & Conrad Armes, D.  (1983). The determination of tinnitus loudness considering the effects of recruitment.  Journal of Speech and Hearing Research, 26(1): 59 72.

You should also cite:

Tyler, R. S., Haskell, G., Gogle, S., & Gehringer, A. (2008).  Establishing a Tinnitus Clinic in Your Practice. Am J Audiol; 17: 25-37.

They highlighted 4 reasons for measuring tinnitus.

1.     Confirm to the patient that their tinnitus is real

2.     Monitor changes

3.     Provide insights into mechanisms

4.     Aid in the fitting of some devices

Tyler, R. S. (1985).  Psychoacoustical measurement of tinnitus for treatment evaluations.  In: E. Myers (Ed.), New Dimensions in Otorhinolaryngology Head and Neck Surgery (455-458). Amsterdam: Elsevier Publishing Co.

Tyler, R. S.  (1992).  The psychophysical measurement of tinnitus.  In:  J-M. Aran & R. Dauman (Eds.), Tinnitus 91 - Proceedings of the Fourth International Tinnitus Seminar (17-26). Amsterdam: Kugler Publications.

Tyler, R. S. (2000).  The psychoacoustical measurement of tinnitus.  In R.S. Tyler (Ed.), Tinnitus Handbook (149-179). San Diego, CA: Singular Publishing Group.

Tyler, R. S., Oleson, J., Noble, W., Coelho, C., & Ji, H. (2007).  Clinical trials for tinnitus: Study populations, designs, measurement variables, and data analysis. Progress in Brain Research, 166: 499-509.

Line 132

Define the duration of the phases.

Line 140

Mate ??

Line 154

Define the protocols so that some can replicate your study, a fundamental of science.

Line 197

Not clear what the 6 month period stands for.

Line 283

Normal hearing..

State how you define this

Normal pure tone thresholds (=< 25 dBHL)

317.  changes over time

Please cite Tyler and Baker, who first documented the wide range of problems experienced by tinnitus sufferers and showed that for most patients, the first 6-9 months is the worst.

Tyler, R.S. and Baker, L.J.  (1983). Difficulties experienced by tinnitus sufferers.  Journal of Speech and Hearing Disorders, 48(2): 150 154.

Line 320

300 Hz.  Are you sure??

Please cite Pan et al.,   whose pitch matching data suggests that most patients do not have a pitch match frequency just below the maximum hearing threshold loss frequency.  The argues against the brain reorganization model in most subgroups of tinnitus.

Pan, T., Tyler, R. S., Ji, H., Coelho, C., Gehringer, A., & Gogel, S. (2009).  The relationship between tinnitus pitch and the audiogram. Int J Audiol.  48 (4): 277-294.

Line 327

dB is not a scale of loudness

Tinnitus sensation level is not a measure of loudness.  You might want to consider referencing:

Tyler, R. S. & Conrad Armes, D.  (1983). The determination of tinnitus loudness considering the effects of recruitment.  Journal of Speech and Hearing Research, 26(1): 59 72.

Line 352

Please cite (Tyler et al., 2014) who focused on the four primary reactions to tinnitus, emotions, hearing, sleep and concentration.  This has been used to measure tinnitus changing daily.

Tyler, R., Ji, H., Perreau, H., Witt, S., Noble, W., & Coelho, C.  (2014). Development and validation of the Tinnitus Primary Function Questionnaire.  American Journal of Audiology, 23, 260–272.

I hope you find my comments helpful.

I would be happy to send reprints if needed.

Rich-tyler@uiowa.edu

Rich Tyler

The University of Iowa

7. PLOS authors have the option to publish the peer review history of their article (what does this mean?). If published, this will include your full peer review and any attached files.

Reviewer #1: No

Reviewer #2: No

---

## [Author Response · Author response to Decision Letter 1]

7 Apr 2021

• Response to Reviewers

More like once again to thank the excellent contributions made about our study. All of them have been taken into account. An additional statistic was performed in order to meet the requests and special attention was given to the writing of the manuscript. Below are the answers to each consideration made.

Reviewer 1:

• The comparison was conducted among groups (difference of the differences) through the non-parametric Mann-Whitney test. This analysis has been added in the "Results", pages 13-16, lines 252-255, 2-271-275, 295-299; and in Appendix 2. 

• THI was best described in terms of composition and analysis in "Methods" ("Questionnaires and exams"), page 7, lines 147-154. This questionnaire was subdivided into three scales - Emotional, Functional and Catastrophic - (Newman et al., 1996), however, a study carried out seeking to analyze the factor structure demonstrated that the THI total score can serve as a robust measure of tinnitus distress and recommended using only the total score both in research and in clinical practice (Baguley; Andreson, 2003). Thus, the authors decided to perform the analysis of the total score without presenting the values by sub-scale. The classification of degrees of tinnitus has been added (Newman et al., 1996): “Results”, Table 1, page 11. As suggested by one of the reviewers, the analysis of the items corresponding to the "lack of control" (questions 8 and 19 of the THI) was added since this represents the areas which may produce the most dramatic effects if changes are observed: “Results”, page 13, lines 244-248 

• The POMS questionnaire was described for composition and analysis in “Methods" ("Questionnaires and exams"), page 8, lines 156-162. The meaning for the abbreviation "TMD" has been added.

• Added a clear description of the study hypothesis: "Introduction", page 04, lines 71-72

• The eligibility criteria were best described: “Methods”, page 5, lines 85-86

• Specification was provided that the value in parentheses referred to the p-value: "Results", page 13, lines 234-239

• Added the description that the values in phase 3 were lower than in phase 1: "Results", page 13, line 239

• The IQR for THI and VAS has been added: "Results”, page 13, line 243 and page 14, lines 250-251

• The test legend used in Table 2 was adjusted: chapter "Results", page 13/14, line 229

• An analysis of the correlation between caffeine consumption and the variables analyzed in the study was added: pages 13-16, lines 252-255, 2-271-275, 295-299; and in Appendix 2. 

• Article writing was revised trying to let it more cohesive and clear

Reviewer 2:

• The description of the placebo was added in the abstract, page 2, line 25

• The choice of a restricted caffeine diet for 24 hours before the study was based on the pharmacology of the substance (presented in the chapter "Methods", page 6, lines 109-114 and in previous studies that sought to investigate the action of caffeine (Hughes, 1991) 

• A more detailed explanation of the Visual Analog scale (VAS) has been added: "Methods" ("Questionnaires and exams"), page 8, lines 153-154.

• Added the explanation of what is the acufenometry: Abstract, page 2, lines 29-30

• The description of what was considered to be a clinically relevant hearing threshold change was obtained in the chapter "Discussion", page 19, lines 376-379

• Added the Psychological Model of Tinnitus: "Introduction", page 3, lines 45-46

• THI was best described in terms of composition and analysis in "Methods" ("Questionnaires and exams"), page 7, lines 147-154. This questionnaire was subdivided into three scales - Emotional, Functional and Catastrophic - (Newman et al., 1996), however, a study carried out seeking to analyze the factor structure demonstrated that the THI total score can serve as a robust measure of tinnitus distress and recommended using only the total score both in research and in clinical practice (Baguley; Andreson, 2003). Thus, the authors decided to perform the analysis of the total score without presenting the values by sub-scale. The classification of degrees of tinnitus has been added (Newman et al., 1996): “Results”, Table 1, page 11. As suggested by one of the reviewers, the analysis of the items corresponding to the "lack of control" (questions 8 and 19 of the THI) was added since this represents the areas which may produce the most dramatic effects if changes are observed: “Results”, page 13, lines 244-248 

• The importance of performing psychoacoustic measurements of tinnitus was presented in the chapter "Introduction", page 3, lines 61-62. The loudness measurement was performed as described by Henry and Zaugg (2005), although the authors recognize the limitations arising from the measures in dB sensation level (Tyler and Conrad, 1983), and this dimension discussed in the "Discussion" , page 19, lines 363-367

• The average duration of the study was added, and the duration in each phase varied according to the individual's response: “Methods”, page 7, line 133

• Added food description: mate herb: “Methods”, page 7, line 144

• The description of the protocols used was revised. Methods Chapter (Questionnaires and exams), pages 7-9

• The 6-month recruitment period was best described: “Results”, page 10, line 214

• The description of normal hearing has been added: “Methods”, page 8, lines 169-170

• The error (3000Hz) has been fixed: Chapter Discussion, page 18, line 355

---

## [Editor Report · Decision Letter 2]

5 Jul 2021

PONE-D-20-30851R2

The effect of caffeine on tinnitus: Randomized triple-blind placebo-controlled clinical trial

PLOS ONE

Dear Dr. Bahmad,

Thank you for submitting your manuscript to PLOS ONE. After careful consideration, we feel that it has merit but does not fully meet PLOS ONE’s publication criteria as it currently stands. Therefore, we invite you to submit a revised version of the manuscript that addresses the points raised during the review process.

We look forward to receiving your revised manuscript.

Kind regards,

Hong-Liang Zhang, M.D., Ph.D.

Academic Editor

PLOS ONE

Additional Editor Comments (if provided):

Line 21

Define what the placebo is in the abstract.

Line 26.

24 hours does not seem long as control…? Explain what this is.

Line 27.

VAS here and elsewhere

I think by VAS you mean magnitude estimation. In psychophysics whenyou are trying to estimate the magnitude of something, it is referred to asMagnitude Estimation. There are several psychophysical procedures thatcan be used to estimate the magnitude. VAS is one. If one is going to usea VAS scale, it is necessary to state the length of the line, what markingsand labels are on the line, and what resolution will be used to convert thesubjects marking into a number.

I prefer to have subjects provide a number from 0 to 100.

Line 29

Most readers will not know what acufenometry is.

Line 35

How do you know what is clinically relevant?

Line 48

Please cite the Psychological Model, proposed by Tyler, Aran and Dauman(1992), suggested that the overall annoyance of the tinnitus was a resultof the 1) tinnitus characteristics and the 2) psychological make up of eachindividual patient. Several parts of the brain will be involved in therepresentation of the tinnitus and of the reactions to the tinnitus. Treatments can be focused on reducing the tinnitus (e.g. pills) or onreducing the reactions to the tinnitus (eg. Counseling )

Tyler, R. S., Aran, J-M., & Dauman, R. (1992). Recent advances intinnitus. Am J Audiol, 1(4): 36-44.

Line 52

The THI actually uses a 3 label category scale. (SS Stevens.. oftenreferred to as the father of human psychophysics)

Please discuss in the results, the sensitivity of the THI has beenchallenged.

Tyler, R.S., Oleson, J., Noble, W., Coelho, C., Ji, H. (2007). Clinical trials fortinnitus: Study populations, designs, measurement variables, and dataanalysis. Progress in Brain Research, 166: 499-509.

Line 53

VAS

I think by VAS you mean magnitude estimation. In psychophysics whenyou are trying to estimate the magnitude of something, it is referred to asMagnitude Estimation. There are several psychophysical procedures thatcan be used to estimate the magnitude. VAS is one. If one is going to usea VAS scale, it is necessary to state the length of the line, what markingsand labels are on the line, and what resolution will be used to convert thesubjects marking into a number.

I prefer to have subjects provide a number from 0 to 100.

Line 60

Measuring tinnitus

The dB sensation level is not a direct measure of loudness.

Please reference

Tyler, R. S. & Conrad Armes, D. (1983). The determination of tinnitusloudness considering the effects of recruitment. Journal of Speech andHearing Research, 26(1): 59 72.

You should also cite:

Tyler, R. S., Haskell, G., Gogle, S., & Gehringer, A. (2008). Establishing aTinnitus Clinic in Your Practice. Am J Audiol; 17: 25-37.

They highlighted 4 reasons for measuring tinnitus.

1. Confirm to the patient that their tinnitus is real

2. Monitor changes

3. Provide insights into mechanisms

4. Aid in the fitting of some devices

Tyler, R. S. (1985). Psychoacoustical measurement of tinnitus fortreatment evaluations. In: E. Myers (Ed.), New Dimensions inOtorhinolaryngology Head and Neck Surgery (455-458). Amsterdam:Elsevier Publishing Co.

Tyler, R. S. (1992). The psychophysical measurement of tinnitus. In: J-M. Aran & R. Dauman (Eds.), Tinnitus 91 - Proceedings of the FourthInternational Tinnitus Seminar (17-26). Amsterdam: Kugler Publications.

Tyler, R. S. (2000). The psychoacoustical measurement of tinnitus. InR.S. Tyler (Ed.), Tinnitus Handbook (149-179). San Diego, CA: SingularPublishing Group.

Tyler, R. S., Oleson, J., Noble, W., Coelho, C., & Ji, H. (2007). Clinical trialsfor tinnitus: Study populations, designs, measurement variables, and dataanalysis. Progress in Brain Research, 166: 499-509.

Line 132

Define the duration of the phases.

Line 140

Mate ??

Line 154

Define the protocols so that some can replicate your study, a fundamentalof science.

Line 197

Not clear what the 6 month period stands for.

---

## [Author Response · Author response to Decision Letter 2]

14 Jul 2021

Response to Reviewers

The last suggested review on the 5th of July contains the same requests as the previous review of the 10th of March 2021.The requests were met in the submitted manuscript and indicated on the document "Response to reviewers" as requested..

Reviewer 2:

• Line 21 -Define what the placebo is in the abstract.

The description of the placebo was added in the abstract, page 2, line 25

• Line 26 - 24 hours does not seem long as control…? Explain what this is.

The choice of a restricted caffeine diet for 24 hours before the study was based on the pharmacology of the substance (presented in the chapter "Methods", page 6, lines 109-114 and in previous studies that sought to investigate the action of caffeine (Hughes, 1991) 

• Line 27 - VAS here and elsewhere. I think by VAS you mean magnitude estimation. In psychophysics when you are trying to estimate the magnitude of something, it is referred to as Magnitude Estimation. There are several psychophysical procedures that can be used to estimate the magnitude. VAS is one. If one is going to use a VAS scale, it is necessary to state the length of the line, what markings and labels are on the line, and what resolution will be used to convert the subjects marking into a number.

I prefer to have subjects provide a number from 0 to 100.

A more detailed explanation of the Visual Analog scale (VAS) has been added: "Methods" ("Questionnaires and exams"), page 8, lines 153-154.

• Line 29 - Most readers will not know what acufenometry is.

Added the explanation of what is the acufenometry: Abstract, page 2, lines 29-30

• Line 35 - How do you know what is clinically relevant?

The description of what was considered to be a clinically relevant hearing threshold change was obtained in the chapter "Discussion", page 19, lines 376-379

• Line 48 - Please cite the Psychological Model, proposed by Tyler, Aran and Dauman(1992), suggested that the overall annoyance of the tinnitus was a resultof the 1) tinnitus characteristics and the 2) psychological make up of eachindividual patient. Several parts of the brain will be involved in therepresentation of the tinnitus and of the reactions to the tinnitus. Treatments can be focused on reducing the tinnitus (e.g. pills) or onreducing the reactions to the tinnitus (eg. Counseling )Tyler, R. S., Aran, J-M., & Dauman, R. (1992). Recent advances intinnitus. Am J Audiol, 1(4): 36-44.

Added the Psychological Model of Tinnitus: "Introduction", page 3, lines 45-46

• Line 52 - The THI actually uses a 3 label category scale. (SS Stevens.. often referred to as the father of human psychophysics)

Please discuss in the results, the sensitivity of the THI has been challenged.

Tyler, R.S., Oleson, J., Noble, W., Coelho, C., Ji, H. (2007). Clinical trials for tinnitus: Study populations, designs, measurement variables, and data analysis. Progress in Brain Research, 166: 499-509.

THI was best described in terms of composition and analysis in "Methods" ("Questionnaires and exams"), page 7, lines 147-154. This questionnaire was subdivided into three scales - Emotional, Functional and Catastrophic - (Newman et al., 1996), however, a study carried out seeking to analyze the factor structure demonstrated that the THI total score can serve as a robust measure of tinnitus distress and recommended using only the total score both in research and in clinical practice (Baguley; Andreson, 2003). Thus, the authors decided to perform the analysis of the total score without presenting the values by sub-scale. The classification of degrees of tinnitus has been added (Newman et al., 1996): “Results”, Table 1, page 11. As suggested by one of the reviewers, the analysis of the items corresponding to the "lack of control" (questions 8 and 19 of the THI) was added since this represents the areas which may produce the most dramatic effects if changes are observed: “Results”, page 13, lines 244-248 

• Line 60 - Measuring tinnitus. The dB sensation level is not a direct measure of loudness. Please reference Tyler, R. S. & Conrad Armes, D. (1983). The determination of tinnitus loudness considering the effects of recruitment. Journal of Speech andHearing Research, 26(1): 59 72. You should also cite: Tyler, R. S., Haskell, G., Gogle, S., & Gehringer, A. (2008). Establishing aTinnitus Clinic in Your Practice. Am J Audiol; 17: 25-37. They highlighted 4 reasons for measuring tinnitus. 1. Confirm to the patient that their tinnitus is real 2. Monitor changes 3. Provide insights into mechanisms 4. Aid in the fitting of some devices Tyler, R. S. (1985). Psychoacoustical measurement of tinnitus fortreatment evaluations. In: E. Myers (Ed.), New Dimensions inOtorhinolaryngology Head and Neck Surgery (455-458). Amsterdam:Elsevier Publishing Co.

Tyler, R. S. (1992). The psychophysical measurement of tinnitus. In: J-M. Aran & R. Dauman (Eds.), Tinnitus 91 - Proceedings of the FourthInternational Tinnitus Seminar (17-26). Amsterdam: Kugler Publications.

Tyler, R. S. (2000). The psychoacoustical measurement of tinnitus. InR.S. Tyler (Ed.), Tinnitus Handbook (149-179). San Diego, CA: SingularPublishing Group.

Tyler, R. S., Oleson, J., Noble, W., Coelho, C., & Ji, H. (2007). Clinical trialsfor tinnitus: Study populations, designs, measurement variables, and dataanalysis. Progress in Brain Research, 166: 499-509.

The importance of performing psychoacoustic measurements of tinnitus was presented in the chapter "Introduction", page 3, lines 61-62. The loudness measurement was performed as described by Henry and Zaugg (2005), although the authors recognize the limitations arising from the measures in dB sensation level (Tyler and Conrad, 1983), and this dimension discussed in the "Discussion", page 19, lines 363-367

• Line 132 - Define the duration of the phases.

The average duration of the study was added, and the duration in each phase varied according to the individual's response: “Methods”, page 7, line 133

• Line 140 - Mate ??

Added food description: mate herb: “Methods”, page 7, line 144

• Line 154 - Define the protocols so that some can replicate your study, a fundamental of science.

The description of the protocols used was revised. Methods Chapter (Questionnaires and exams), pages 7-9

• Line 197 - Not clear what the 6 month period stands for.

The 6-month recruitment period was best described: “Results”, page 10, line 214

• The description of normal hearing has been added: “Methods”, page 8, lines 169-170

• The error (3000Hz) has been fixed: Chapter Discussion, page 18, line 355

---

## [Editor Report · Decision Letter 3]

4 Aug 2021

The effect of caffeine on tinnitus: Randomized triple-blind placebo-controlled clinical trial

PONE-D-20-30851R3

Dear Dr. Bahmad,

We’re pleased to inform you that your manuscript has been judged scientifically suitable for publication and will be formally accepted for publication once it meets all outstanding technical requirements.

Kind regards,

Hong-Liang Zhang, M.D., Ph.D.

Academic Editor

PLOS ONE

Additional Editor Comments (optional):

All of the reviewers' concerns have been addressed. No further revisions are required.
---

## [Editor Report · Acceptance letter]

9 Sep 2021

PONE-D-20-30851R3 

The effect of caffeine on tinnitus: Randomized triple-blind placebo-controlled clinical trial
Short title: The effect of caffeine on tinnitus 

Dear Dr. Bahmad Jr:

I'm pleased to inform you that your manuscript has been deemed suitable for publication in PLOS ONE. Congratulations! Your manuscript is now with our production department. 

Kind regards, 

on behalf of

Dr. Hong-Liang Zhang 

Academic Editor

PLOS ONE